# Neuronal Death Caused by HMGB1-Evoked via Inflammasomes from Thrombin-Activated Microglia Cells

**DOI:** 10.3390/ijms241612664

**Published:** 2023-08-11

**Authors:** Meei-Ling Sheu, Liang-Yi Pan, Cheng-Ning Yang, Jason Sheehan, Liang-Yu Pan, Weir-Chiang You, Chien-Chia Wang, Hong-Shiu Chen, Hung-Chuan Pan

**Affiliations:** 1Institute of Biomedical Sciences, National Chung-Hsing University, Taichung 40227, Taiwan; mlsheu@nchu.edu.tw; 2Department of Medical Research, Taichung Veterans General Hospital, Taichung 40210, Taiwan; 3Ph.D. Program in Translational Medicine, Rong Hsing Research Center for Translational Medicine, National Chung Hsing University, Taichung 40227, Taiwan; 4Faculty of Medicine, Kaohsiung Medical University, Kaohsiung 80708, Taiwan; pan0911606850@gmail.com; 5Department of Dentistry, School of Dentistry, College of Medicine, National Taiwan University, Taipei 106319, Taiwan; cnyang880@yahoo.com.tw; 6Department of Neurosurgery, University of Virginia, Charlottesville, VA 22904, USA; jps2f@hscmail.mcc.virginia.edu; 7Faculty of Medicine, Poznan University of Medical Sciences, 61-701 Poznań, Poland; mattpan9009@gmail.com; 8Department of Radiation Oncology, Taichung Veterans General Hospital, Taichung 40210, Taiwan; bigjohnyou@gmail.com; 9Department of Life Sciences, National Central University, Taoyuan 32001, Taiwan; superdukewang@gamil.com; 10Department of Neurosurgery, Tungs’ Taichung Metro-Harbor Hospital, Taichung 40210, Taiwan; hungshinchen@gmail.com; 11Department of Neurosurgery, Taichung Veterans General Hospital, Taichung 40210, Taiwan

**Keywords:** thrombin, inflammasome, endoplasmic retinaculum stress, high-mobility group box chromosomal protein 1(HMGB-1)

## Abstract

Microglial cells are a macrophage-like cell type residing within the CNS. These cells evoke pro-inflammatory responses following thrombin-induced brain damage. Inflammasomes, which are large caspase-1-activating protein complexes, play a critical role in mediating the extracellular release of HMGB1 in activated immune cells. The exact role of inflammasomes in microglia activated by thrombin remains unclear, particularly as it relates to the downstream functions of HMGB1. After receiving microinjections of thrombin, Sprague Dawley rats of 200 to 250 gm were studied in terms of behaviors and immunohistochemical staining. Primary culture of microglia cells and BV-2 cells were used for the assessment of signal pathways. In a water maze test and novel object recognition analysis, microinjections of thrombin impaired rats’ short-term and long-term memory, and such detrimental effects were alleviated by injecting anti-HMGB-1 antibodies. After thrombin microinjections, the increased oxidative stress of neurons was aggravated by HMGB1 injections but attenuated by anti-HMGB-1 antibodies. Such responses occurred in parallel with the volume of activated microglia cells, as well as their expressions of HMGB-1, IL-1β, IL-18, and caspase-I. In primary microglia cells and BV-2 cell lines, thrombin also induced NO release and mRNA expressions of iNOS, IL-1β, IL-18, and activated caspase-I. HMGB-1 aggravated these responses, which were abolished by anti-HMGB-1 antibodies. In conclusion, thrombin induced microglia activation through triggering inflammasomes to release HMGB1, contributing to neuronal death. Such an action was counteracted by the anti-HMGB-1 antibodies. The refinement of HMGB-1 modulated the neuro-inflammatory response, which was attenuated in thrombin-associated neurodegenerative disorder.

## 1. Introduction

Neuro-inflammation has been implicated in neurodegenerative disorders for decades. However, the exact timing and mechanism of the inflammation process in neurodegenerative disorders remains poorly understood [1]. Specifically, whether inflammation is a trigger that results in neurodegeneration remains to be clarified. Growing evidence supports the role of microglial activation in neuronal damage leading to neurodegeneration [2,3,4]. Microglia, the tissue macrophages of the brain, under healthy conditions, are a resting phenotype characterized by a ramified morphology, and they extend their fine processes to scan the environment [5,6]. In response to a homeostatic disturbance, microglia rapidly change their phenotype, thereby contributing to processes such as inflammation, stroke, trauma, tissue remodeling, and neurogenesis [4,7,8,9,10].

Recently, mounting evidence has shown that certain elements of the coagulation cascade initiated and propagated the inflammation response in the central nervous system. The potentially important protein related to both coagulation and inflammation is thrombin [11,12]. Thrombin is well known as the serine protease in the blood coagulation cascade [13], and it also functions as a potent signaling molecule that widely regulates physiologic and pathogenic responses across cell populations and tissues [1,4,14,15]. Following brain injury and other cerebrovascular damage, the activation of prothrombin and the leakage of active thrombin into brain parenchyma cause overactive inflammatory responses, often leading to irreversible neuronal damage. Thrombin not only activates endothelial cells and induces leukocyte infiltration and edema but also causes microglia to propagate focal inflammation and potentially neurotoxic effects [4,14,15].

NOD (nucleotide-binding oligomerization domain)-like receptors (NLRs) are a family of intracellular sensors of microbial motifs and ‘danger signals’. NLRs are crucial components of the innate immune responses and inflammation [16,17,18]. NLRs sense nonmicrobial danger signals and form large cytoplasmic complexes called inflammasomes that link the sensing of microbial products and metabolic stress to the proteolytic activation of the proinflammatory cytokines of IL-1β and IL-18 by activating caspase-1 [19,20,21]. The NLRP3 (NLR family pyrin domain containing 3) inflammasome is implicated in inflammatory conditions, such as trauma, atherogenesis, and bacterial infection. Activated NLRP3 inflammasomes sense a variety of diverse molecules, and they are associated with inflammatory disorders [22,23,24].

High-mobility group box chromosomal protein 1 (HMGB-1), a DNA (deoxyribonucleic acid)-binding protein, has been recognized as a novel candidate in a specific upstream pro-inflammation pathway after brain injury [25,26,27]. Minocycline, a semisynthetic tetracycline antibiotic, has a palliative action with a new therapeutic potential for treating post-ischemic injury via an HMGB1 inhibitor action [28,29,30,31,32,33,34,35,36,37,38,39]. Emerging evidence has shown that activated microglia express HMGB1 activation in cerebral ischemia, leading to marked neurological impairments. Paradoxically, HMGB1 is involved in silencing innate immunity at the cellular and molecular levels, and suppresses inflammation. On the contrary, HMGB1 promotes neurogenesis and participates in brain tissue remodeling [31,32,33,34]. HMGB1 is a prototypical danger signal that regulates inflammatory and repair responses. With infection, injury and sterile inflammation, HMGB1 is passively released from damaged cells or actively secreted from activated immune cells. Inflammasomes, large caspase-1-activating protein complexes, play a critical role in mediating the extracellular release of HMGB1 from activated and infected immune cells [35,36,37]. However, the role of HMGB1 regulated by the inflammasome in thrombin activated microglia cells either in vitro or in vivo has not been well defined. 

In this study of inflammatory processes, we investigated thrombin-induced microglia activation and focused on the involvement of inflammasomes related to the cascade of HMGB1 production. These processes contributed to neuronal death and were either exaggerated by HMGB1 or countered by the administration anti-HMGB-antibodies. Through a better understanding of the role of HMGB-1 in the suppression of neuro-inflammatory reactions, the refined anti-HMGB-1 antibodies could shed light on treating thrombin-associated neuro-inflammation diseases. 

## 2. Results

### 2.1. Deterioration of Short- and Long-Term Memory by Co-Administration of Thrombin and HGMB1 and Its Alleviation by Anti-HGMB1 Antibodies (NA-HMGB1) 

To investigate effects of HMGB1 on thrombin-activated microglia related to the oxidative stress of neurons, we stereotactically micro-injected a mixture of thrombin with either LPS or HMGB1 along with anti-HMGB1 antibody at the hippocampus CA3 region. The LPS group served as the positive control. In the water maze test, we measured parameters, like swimming speed, latency to the target, % of time in target quadrant, % of time in target quadrant in extinction, number of times crossing over the target, and % of time in reverse target quadrant. We found that either LPS or thrombin exerted a significant memory impairment compared to the PBS control. The co-administration of LPS and thrombin produced the worst memory impairment, compared to LPS or thrombin alone. The deterioration of memory caused by thrombin was aggravated by the addition of HMGB1, and the effect was attenuated by the anti-HMGB1 antibody (Figure 1A–F). For more details on the comparison of the different groups as a function of time, the plots illustrated in bar graph form are shown in Appendix A. Similar results or trends were found with the novel object recognition in the short- and long-term memory tests when compared to the water maze test (Figure 2A,B).

### 2.2. Detrimental Effects on Neurons Exerted by Activated-Microglia-Paralleled Behavioral Results Especially in Memory Impairments

An increased expression of 8-oxo-dG was observed in animals subjected to a microinjection of either LPS or thrombin. A marked expression of 8-oxo-dG was found in the group with the co-administration of LPS and thrombin. HGMB1 synergistically increased 8-oxo-dG expressions in thrombin-activated microglia cells, but this change was counteracted by the administration of NA- HMGB1 (Figure 3A). The number of activated microglia cells reflected the status of the inflammatory response. Either LPS or thrombin induced a substantially greater number of microglia cells. The synergistic effect was observed with the combined administration of LPS and thrombin, and the effect was abolished by administration of the neutralized antibody of HMGB1. HMGB1 itself had only a mild effect on the activation of microglia, but it significantly increased the number of thrombin-activated microglia cells (Figure 3B). To investigate if the activation of microglia by thrombin was correlated to the cascades of inflammasomes, we subjected the activated microglia to the assessment of the co-localization of HMGB1, IL-1β, IL-18, and caspase-I with microglia cells. We found significant expressions of HMGB1, IL-1β, IL-18, and caspase-I in the groups receiving combined treatments of thrombin and LPS compared to groups receiving either thrombin or LPS. HMGB1 synergistically increased the endogenous expressions of HMGB1, IL-1β, IL-18, and caspase-I in microglia activated by thrombin, while these expressions were attenuated by NA-HMGB1 (Figure 3C–F). In further quantitative analysis of the number of activated microglia cells and 8-oxo-dG, results further confirmed the immunohistochemical findings (Figure 4A,B). The obtained tissue in the region of hippocampus were subjected to Western blot analysis. The intensity of immunohistochemistry staining was also quantified (Appendix A), and the analysis showed the same expression pattern consistent with the representative Western blots. 

### 2.3. Thrombin-Triggered Activation of Inflammasomes in Microglia Cells

Nitrite formation and iNOS expression are hallmarks of microglia activation. At a dose of 10 units of thrombin, only minimal NO production was observed. However, with dosages >25 units, we found greater NO expression. LPS synergistically enhanced NO production even with thrombin at a dose of 1 unit, and escalated responses were found with doses ranging from 5 to 25 units (Figure 5A). The iNOS were induced in microglia cells subjected to either thrombin or LPS treatment. However, LPS exerted a synergistic effect on thrombin in augmenting iNOS expression (Figure 5B,C). The cell morphological alteration and cell viability were shown in Appendix A. The MTT assay mirrored the trend of NO production. 

To investigate the involvement of inflammasomes in the above thrombin activation, we determined the mRNA expressions of IL-1β and IL-18 in microglia cells. Thrombin had activated a marked expression of IL-1β with a progressive increase in this response from 1 to 50 units thrombin, whereas LPS served as the positive control. LPS synergistically increased the mRNA expressions of IL-1β in microglia cells following the dosage of thrombin from 1 to 10 units (Figure 6A). The escalated dosage of thrombin from 1 to 50 units exerted the significant expression of IL-18 mRNA, and LPS was used as a positive control. LPS synergistically increased mRNA expressions of IL-18 in microglia cells following the dosage of thrombin from 1 to 10 units (Figure 6B).

In the supernatant of microglia cells subjected to thrombin stimulation, the expression trend of IL-1β and IL-18 paralleled the results of mRNA expression (Figure 7A,B). The whole cell lysate analysis in the Western blot analysis revealed that thrombin had triggered expressions of IL-1β and IL-18 in microglia, and there was an additional effect with LPS administration (Figure 7C,D). The caspase-I expression, especially at its P20 form, showed phenomena similar to those observed with IL-1β and IL-18 (Figure 7E). The quantitative analyses with IL-1β, IL-18, and caspase-I further confirmed the above findings (Figure 7F–H).

### 2.4. Release of HMGB1 Triggered by Thrombin with a Parallel Upregulation of IL-1β and IL 18

There was a mildly increased secretion of HMGB1 in microglia subjected to thrombin stimulation at doses as high as 10 units. Thrombin at doses up to 25 units caused a significant secretion of HMGB1 in microglia cells. Furthermore, LPS had an augmented effect on thrombin in escalating HMGB1 secretion, even at doses as low as 10 units, and the response reached a plateau at a dose of 20 units. This augmentation was strongly attenuated by the neutralized antibodies of HMGB1 (Figure 8A). The increased secretion of IL-1β was induced by thrombin at a dose of 20 units, and this response was also escalated by HMGB1 (Figure 8B). Similar phenomena were also observed on IL-18 secretion. Increased secretion of IL-18 was induced by 20 units of thrombin, and this response was also accelerated by HMGB1 (Figure 8C).

### 2.5. HMGB1 Released Regulated by Cathepsin B Activity in Microglia Triggered by Thrombin

Cathepsin B is required for inflammasome activation in immune cells through NLRP3 interaction. To investigate the possibility of the inflammasome involvement in the pathway of cathepsin B, we measured the cathepsin B activity of microglia cells after thrombin stimulation. In LPS-pretreated microglia cells, after further additions of thrombin, we found roughly a 2.5-fold increase in cathepsin B activity at a thrombin dose of 25 units, and the increase went up to three fold at a dose of 50 units (Figure 9A). Cathepsin B inhibitors, at a dose of 5 unit, reduced HMGB1 activity in thrombin-activated microglia cells, and they completely abolished the activity at a dose of 10 units (Figure 9B). The results indicated the involvement of cathepsin B in the release of HGMB1 triggered by thrombin.

## 3. Discussion

In the current study, we have demonstrated that thrombin-treated microglia showed a downstream activation of inflammasomes, thereby contributing to HMGB1 secretion. The thrombin-activated expressions of IL-1β and IL-18, coupled with the activation of caspase-I, were augmented by HMGB1 but abolished by anti-HMGB1 antibody. These effects were highly correlated with changes in neurobehavior and neuronal survival. Hence, modulating HMGB1 in the thrombin-activated inflammatory responses can play a potential role in the reversal of a neurodegenerative disorder.

In the brain, iNOS is expressed in glial cells, such as astrocytes and microglia [38,39]. The expression of iNOS is induced or stimulated, typically by proinflammatory cytokines and/or bacterial lipopolysaccharide (LPS) [40,41]. An overexpression or dysregulation of iNOS may be toxic, as it is associated with a number of disorders, like septic shock, cardiac dysfunction, pain, diabetes, and cancer [40]. NO produced by activated microglia is toxic to neighboring cells [42,43,44]. The expression of iNOS is markedly increased in microglia cells after thrombin treatment both in vitro and in vivo [45,46]. In this study, we found that LPS increased iNOS production, and it was synergetic when combined with thrombin. Our results indicated their successful induction of inflammatory responses, which was consistent with previous studies.

NLRs likely detect non-microbial danger signals and form large cytoplasmic complexes called inflammasomes that link the sensing of microbial products and metabolic stress to the proteolytic activation of proinflammatory cytokines, such as IL-1β and IL-18, through caspase-1 activation [19,20,21]. NLRP3 inflammasomes contain NLRP3, apoptosis-associated speck-like protein (ASC) and caspase-1. ASC regulates interactions between NLRP3 and caspase-1. It senses a variety of danger signals, and it activates caspase-1 and promotes the release of IL-1β, IL-18, and IL-33 [40,41]. NLRP3 is located mostly in microglia [47,48,49]. The activation of NLRP 3 by thrombin in BV-2 microglia cells is a subsequent inflammatory response, and it is attenuated by various pharmacological agents [50,51,52]. In this study, primary microglia cells triggered by thrombin increased the production of the IL-1β and IL-18 via caspase-I activation. Our finding of its simultaneous detection with pro-caspase I enzyme is consistent with the activation of caspase-I in triggering the downstream response of IL-1β and IL-18.

In lipopolysaccharide (LPS)-induced endotoxemia and sepsis, HMGB1 is an extracellularly released mediator in both inflammatory and repair responses [53]. In animal models of endotoxemia and sepsis, passive immunization with HMGB1-neutralizing antibodies prevents organ damage [53,54,55,56]. HMGB1 induces the recruitment of inflammatory cells, and it contributes to dendritic cell maturation and the proliferation of activated T cells [57,58]. Therefore, extracellular HMGB1 is associated with both inflammatory and repair responses in infectious diseases, trauma, and autoimmune disorders. HMGB1 represents a danger-sensing molecule through its immune receptors. During infection or injury, activated immune cells and damaged cells release HMGB1 into the extracellular space, where HMGB1 functions as a pro-inflammatory mediator and plays an important role in the pathogenesis of inflammatory diseases [59,60].

Double-stranded RNA-dependent kinase, an intracellular danger-sensing molecule, physically interacts with inflammasomes, and it is important for inflammasome activation and HMGB1 release. Together, the previous studies described above not only better define novel mechanisms of HMGB1 release during inflammation, but they also provide potential therapeutic targets to treat HMGB1-related inflammatory diseases [37]. In this study, we had found that thrombin activated inflammasomes, which was determined by pro-caspase I activation followed by an increase in IL-1β and IL-18 expression. Such responses were aggravated by HMGB1 but attenuated by HGMB1-neutralizing antibodies. Such results are consistent with thrombin triggering the activation of inflammasomes with HMGB1 release, contributing to neuronal death. This response was abolished by the HMGB1 neutralizing antibodies.

An intriguing question that has attracted considerable attention is how HMGB1 travels from the nucleus to the extracellular milieu given the absence of a classical secretion signal since such signaling peptides are typically required for transporting secretory proteins through the endoplasmic reticulum (ER) lumen to the extracellular space via Golgi-derived secretory vesicles [61,62]. However, certain cytoplasmic and nuclear proteins that lack classical secretion-signal peptides can still reach the extracellular milieu through ER- and Golgi-independent pathways [63]. Indeed, HMGB1 was shown to be released from LPS-activated monocytes and macrophage cell lines, a finding that is consistent with its role as a prototypical alarmin [36]. Cathepsin B is synthesized at the rough endoplasmic reticulum as a pre-proenzyme that consists of 339 amino acids with a signal peptide of 17 amino acids [64,65]. Procathepsin B of 43/46 kDa is then transported to the Golgi apparatus, where cathepsin B is formed. Cathepsin B may enhance the activity of other proteases, including matrix metalloproteinase, urokinase (serine protease urokinase plasminogen activator), and cathepsin D [66,67], as well as the mannose-6 phosphate- and mannose-6 phosphate-receptor-mediated transport of pro-cathepsin from the trans-Golgi network to endo/lysosome. In lysosomes, pro-Cathepsin is further processed via autocatalysis into a mature two-chain form composed of an N-terminal light chain and a C-terminal heavy chain [68,69]. Thus, cathepsin B likely has essential roles in proteolyzing the extracellular matrix, disrupting intercellular communication, and reducing the expression of protease inhibitors [70]. The thrombin-activation of cathepsin B, which we found in this study, was related to the release of HMGB1. This process is likely related to transporting secretory proteins through the endoplasmic reticulum (ER) lumen to the extracellular space in Golgi-derived secretory vesicles through the means reported in the literature [61,62].

The hippocampus is part of the limbic system. It plays a critical role in emotion, fear, learning, and memory [71,72,73,74]. The major functions of the CA1 sector and CA3 sector of the hippocampus proper are spatial and contextual memory, respectively [75,76]. Neurotoxic lesions or the pharmacological inactivation of these areas impair the encoding and retrieval of contextual memory [77,78]. CA3 sector supports processes involved in spatial pattern association, spatial pattern completion, novelty detection, and short-term memory. The CA1 sector is responsible for the temporal pattern completion and intermediate-term memory [79]. In some aspects, the CA1 sector plays a role as a ‘novelty’ detector since it appears to identify the mismatches between the set of inputs from the entorhinal cortex (regarding the current situation) and those from the CA3 sector [10]. In a water maze test, recent and remote memory are similarly impaired after damaging the hippocampus [9]. In this study, thrombin triggered neuronal death, which was aggravated by the administration of HMGB1, but thrombin was attenuated by HMGB1-neutralizing antibodies. The extent of neuronal death was highly correlated to the neuro-behavior as demonstrated in the water maze and EthoVison tests. In general, these results support a possible role of HMGB1 in thrombin-related neurodegenerative disorder.

## 4. Materials and Methods

### 4.1. Cell Culture

Microglia cells were cultured from cerebral cortices of Sprague Dawley rats as previously described [80,81,82]. In brief, cell cultures were prepared from 17-day-old fetal brains through mechanically triturating the neocortex. Dissociated cells were plated on a 12 mm round plastic coverslip in 24-well plates at a density of 1.5 × 10^5^ cells/coverslip. Plating media consisted of Eagle’s minimal essential medium (MEM, Earle’s salts, supplied glutamine-free) supplemented with 5% horse serum, 5% fetal bovine serum (FBS), 21 mM glucose, 26.5 mM bicarbonate, and 2 mM L-glutamine. For neuron-rich cultures (95% pure), 2.5 μM arabinoside C was added between the 2nd and 3rd day in vitro (DIV 2–3). Cortices were triturated into single cells in DMEM containing 10% fetal bovine serum and plated into 75 cm2 T-flasks (0.5 hemispheres per flask) for 2 weeks. Microglia cells were detached from the flasks after mild shaking, and a nylon mesh was applied to remove astrocytes and cell clumps. Cells were plated into 24-well plates. Plates were washed 1 h later with the medium to remove unattached cells. In some experiments, an immortalized microglial cells of cell line BV-2 were cultured and maintained in DMEM medium containing 10% heat-inactivated low-endotoxin FBS (Life Technologies, Carlsbad, CA, USA) and streptomycin/penicillin (Life Technologies) in a humidified atmosphere containing 5% CO_2_.

### 4.2. Immunoblotting Analysis

Proteins levels were determined according to methods reported previously [80,81,82]. In brief, protein (60 μg) was separated by SDS-PAGE, electrophoretically transferred to nitrocellulose membranes, and blocked for 1 h in phosphate-buffered saline containing Tween 20 (0.1%) and nonfat milk (5%). Blots were incubated for 1 h with iNOS (# 06-573; Rabbit Polyclonal; 1:200; Merck Millipore) and caspase-I (# No:bs-0169R; Rabbit Polyclonal; 1:200; Bioss, Woburn, MA, USA), and β actin (# A2228; Mouse Monoclonal; 1:200; Sigma-Aldrich, St. Louis, MO, USA) antibodies were incubated with horseradish peroxidase-conjugated secondary antibody, which were developed using ECL Western blotting reagents. The intensity of protein bands was determined with a computerized image analysis system (IS1000, Alpha Innotech Corporation, Santa Clara, CA, USA).

### 4.3. ELISA

HMGB-1, IL-1, and IL-18 were detected using commercial kits as follows: ELISA kits for HMGB-1 (Cat# No: CSB-E08225m; abs, ASIA Bioscience Co. Ltd., Woburn, MA, USA), IL-1β (Catalog No:559630; BD Biosciences, San Diego, CA, USA), and IL-18 (Catalog No: 50073-MNCE; abs, ASIA Bioscience Co. Ltd.), respectively. Equal amounts of protein were used for ELISA according to manufacturer’s instructions.

### 4.4. Stereotaxic Surgery and Reagents Microinjection

The method of modified stereotaxic microinjection was developed in our laboratory and published [9]. Sprague Dawley rats (230–250 gm) were first anesthetized with chloral hydrate (400 mg/kg i.p.) and then fixed in a stereotaxic apparatus. We stereotactically injected into the right cerebral cortex (AP + 1.4 mm, ML −2.0 mm, DV −2.0 mm from bregma) either PBS, LPS (L) (# L4391, Sigma, St. Louis, MO, USA), thrombin (T) (# T4648, Sigma-Aldrich, St. Louis, MO, USA), HMGB1 (#1690-HMB-050, R&D, Minneapolis, MN, USA), HMGB1-neutralized antibody (NA-HMGB1) (# GTX629400, GeneTex, Irvine, CA, USA), or any combination of the above 2 or 3 items). In this study, we applied injections at 0.5 μL/min with a 26-gauge Hamilton syringe attached to an automated pump and left in situ for an additional 5 min to avoid reflux along the injection tract. The injections were either 5 μg of LPS, 20 units of thrombin, HMGB1 (20 μg), or HMB1 neutralized antibody (NA-HMGB1) (20 μg) dissolved in 5 μL of phosphate-buffered saline. PBS 5 μL served as the control injection.

### 4.5. Water Maze Test

The method involving the modified water maze was reported in our previous study [9]. In brief, a 1.5 m diameter, 45 cm deep Morris water maze tank was filled with water to a depth of 26.5 cm. Water temperature was kept at 26 ± 2 °C. A circular platform, 25 cm high and 12 cm in diameter was placed into the tank at a fixed location at the center of one of four imaginary quadrants. Approximately 1.5 L of milk was added to make the water opaque. Immediately prior to behavior testing, rats were allowed to adapt for 10–15 min to the experimental area (to the white light). Each rat was given 1 min to stay on the platform before undergoing 3 swim training trials. During each training trial the rat was removed from the stand and released facing the platform at a distance of 12–18 inches. The rat was guided by the experimenter to swim toward the platform. A rest period of 30 s was followed by 12 test trials: two sets each of 6 trials, with 4 trials from each of the 3 starting positions and a rest period of 30–45 min (in a small warm cage in a different room) separating the two test periods. During the test trial, each rat was released into the water at one of the 3 different starting positions facing the wall and presumably using distal cues (e.g., cabinets, doors, posters, etc.) to navigate to the platform. The starting positions were all at equal distance from the platform, located immediately adjacent to the wall at the respective center of the 3 quadrants excluding the platform. The rat was allowed to swim for up to 1 min to locate the platform. If it failed the test, escape was assisted. Rats were given a 60 s inter-trial interval to stay on the platform. Distance traveled and escape latency were measured for each trial. Swim speeds (cm/s) were calculated by dividing the distance traveled by the escape latency. Rats were subjected to analysis one day before stereotactic injection (−1) and then after stereotactic injection, on days 3, 7, 14, 21, and 28.

### 4.6. EthoVison XT with Novel Object Test

The test of novel objects was used to assess visual memory of rodents according to published procedure [83]. Animals were first exposed to two identical visual objects, and then one object was replaced by a new (novel) object. The time spent in exploring each of the two objects was measured. This novel object test consisted of three phases: First phase: the animal was placed in the empty arena for 10 min to habituate to the environment. After 10 min, the animal was returned to its home cage. Second phase: after staying 15 min in its home cage, the animal was put back in the arena, where the two objects were lastly placed. The animal was put at a position midway between the two objects, with its nose pointing towards the wall. The animal was tracked for 3 min and then returned to its home cage. Third phase: one of the two objects was replaced with a novel object and the animal was put back in the arena again. Animal tracking was performed for 3 min. For the assessment of its short- and long-term memory, the study was conducted 1 h and 24 h after the change of novel object.

### 4.7. Immunohistochemistry

Animals were perfused transcardially with saline solution containing 0.5% sodium nitrate and heparin (10 U/mL) followed by 4% paraformaldehyde in 0.1 M phosphate buffer (PB). Brains were removed and post-fixed for 1 h, washed in 0.1 M PB and then immersed in 30% sucrose solution until they sank. Tissues were sectioned on a sliding microtome at 40 μm thickness, and every 6th serial section was processed for immunostaining. In brief, brain sections were incubated in 0.2% Triton X-100 for 30 min, rinsed twice in PBS with 0.5% bovine serum albumin (BSA), and finally incubated overnight at room temperature with primary antibodies. The various primary antibodies used were against CD11b (#1457; 1:400; Serotec, Oxford, UK), IL-1β (#SC-7884; 1:100; Santa Cruz, Santa Cruz, CA, USA), IL-18 (#SC-7954; 1:100; Santa Cruz), caspase -I (#bs-0169R; 1:200; Bioss), and HMGB1 (#3939S, 1:400; Cell Signaling Technology) for microglia, and NeuN (#MAB377B; 1:500; Chemicon, Tokyo, Japan) and 8-oxo-dG (#4354-MC-050;1:500; R&D, Minneapolis, MN, USA) for neurons. AF 488 donkey anti-mouse IgG and AF594 donkey anti-rabbit (1:200; Invitrogen, Waltham, MA, USA) were used for secondary antibodies, and sections were viewed through an immunofluorescence microscope. Five consecutive sections of interest were chosen and measured with the IS 1000 image analysis system according to procedures we described previously [9]. The numbers of cells (CD11b-positive) were counted in 20 square areas randomly selected from a 100-square ocular grid. Six samples in each immunohistochemical staining were analyzed. Fluorescent imaging was performed using an Olympus BX40 Research Microscope, using the same laser power and exposure time. Images were quantitatively analyzed using software ImageJ V1.52 and UN-SCAN-IT gelTM (Gel & Graph Digitizing Software Version 6.1). The counts of CD11b were analyzed using UN-SCANT-IT gel TM software version 6.1 and the others were analyzed using ImageJ software V1.52. In brief, the imaging of immunohistochemistry staining was imported. Next, the original image is then split into 3 single-colored images, and the unwanted image was closed. The desired images were saves as TIFF files. In the threshold window, pixels that were within the red box were selected, and a dark background was also selected. The bar was dragged as needed to modify the selected area. A binary image was created and fluorescence intensity and signal were determined. All the stained pixels were converted to black, and all the non-stained pixels were converted to white. The number of black pixels then indicated the area showing positive staining. The data were shown in the number of back pixel/mm^2^.

### 4.8. Nitrite/Nitrate Assay

Concentrations of nitrite in the cell culture supernatant were determined using the nitrite/nitrate colorimetric assay kit (R&D Systems, Minneapolis, MN, USA). Nitrite, an end-product of NO oxidation, was used as an indicator of NO production. Nitrite in the conditioned medium was determined using the Griess reagent. Absorbency was determined at 550 nm using a thermos-microplate reader (Molecular Devices, San Jose, CA, USA) [82].

### 4.9. Cathepsin B Activity Assay Kit

Microglia cells (1–5 × 10^6^ cells) were collected by centrifugation. Cells were lysed in 50 μL of chilled CB Cell Lysis Buffer and incubated on ice for 10 min before micro-centrifuged at top speed for 5 min. Supernatants were transferred to a 96-well plate, in, with 50 μL of each cell lysate, and a similar process was also followed for the uninduced control cell lysate. Assay procedures were conducted according to manufacturer’s recommendation: 50 μL of CB reaction buffer was added to each induced and control well, followed by 2 μL of 10 mM CB substrate Ac-RR-AFC. Samples were incubated at 37 °C for 1–2 h. Fluorescence (RFU) was determined at 505 nm.

### 4.10. Statistical Analyses

Experimental values were presented as mean ± S.E.M. All analyses were performed by ANOVA followed by a Fisher’s least significant difference test. Statistical significance was set at *p* < 0.05.

## 5. Conclusions

Thrombin may induce microglia-activation-triggered inflammasomes to produce HMGB-1 downstream and resulted in detrimental effects to nearby neurons. Such effects were attenuated by HMGB1-neutralizing antibodies. The refinement of HMGB-1 expression may be a useful tool for modulating neuro-inflammatory responses, thereby attenuating a thrombin-associated central nervous system degenerative disorder cascade.

## Figures and Tables

**Figure 1 ijms-24-12664-f001:**
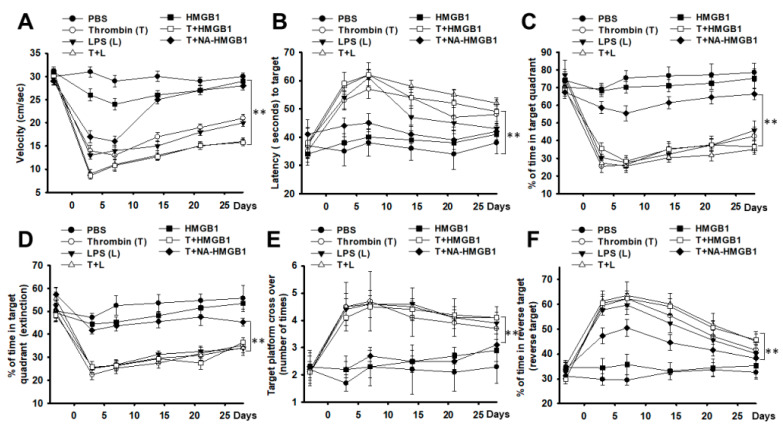
Representative water maze tests in different treatment groups. (**A**) Quantitative analysis of swimming speed in different treatment groups. (**B**) Quantitative analysis of swimming latency to the target in different treatment groups. (**C**) Quantitative analysis of percentage of time in quadrant target in different treatment groups. (**D**) Quantitative analysis of percentage of time in quadrant target at the extinction in different treatment groups. (**E**) Quantitative analysis of times of cross over to the target at the extinction in different treatment groups. (**F**) Quantitative analysis of the percentage in the reverse target in different treatment groups. PBS, L, T, T + L, HMGB1, T + HMGB1, T + NA-HMGB1: see text. N = 6, ** *p* < 0.01.

**Figure 2 ijms-24-12664-f002:**
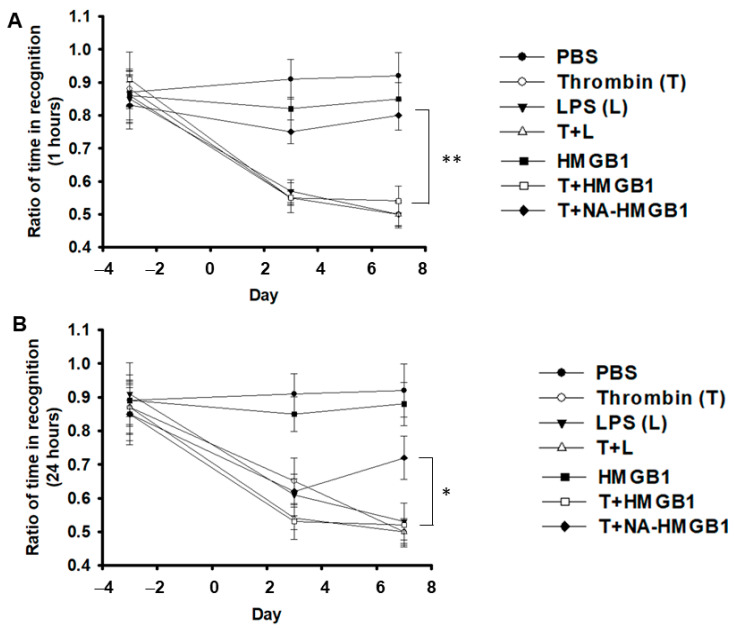
Representative novel object recognition tests in different treatment group. (**A**) Quantitative analysis of percentage of time in novel object recognition one hour after alteration. (**B**) Quantitative analysis of percentage of time in novel object recognition 24 h after alteration. PBS, L, T, T + L, HMGB1, T+ HMGB1, T + NA-HMGB1: see text. N = 6; * *p* < 0.05; ** *p* < 0.01.

**Figure 3 ijms-24-12664-f003:**
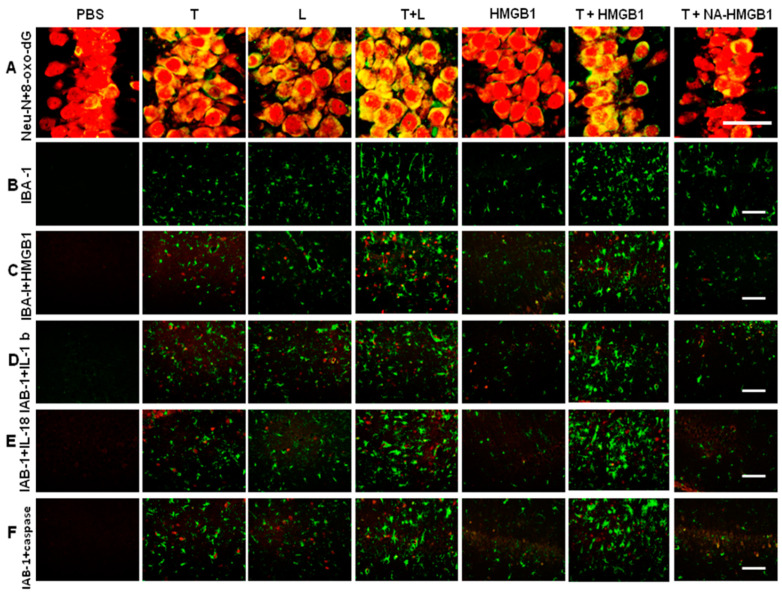
Illustration of oxidized status of neuron and activated microglia cells subjected to different treatment groups 7 days after operation. (**A**) Merged imaging of Neu-N (red) and 8-oxo-dG (green) over the CA3 region of the hippocampus in different treatment groups. (**B**) Illustration of IBA1 (green) surrounding the CA3 region in different treatment groups. (**C**) Merged imaging of IBA1 (green) and HMGB1 (red) in the different treatment groups. (**D**) Merged imaging of IBA1 (green) and IL-1β (red) in the different treatment groups. (**E**) Merged imaging of IBA1 (green) and IL-18 (red) in the different treatment groups. (**F**) Merged imaging of IBA1 (green) and caspase I (red) in the different treatment groups. T = thrombin; L = LPS; L + T = co-injection of LPS and thrombin; PBS, L, T, T + L, HMGB1, T+ HMGB1, T + NA-HMGB1: see text. Bar length = 100 μm.

**Figure 4 ijms-24-12664-f004:**
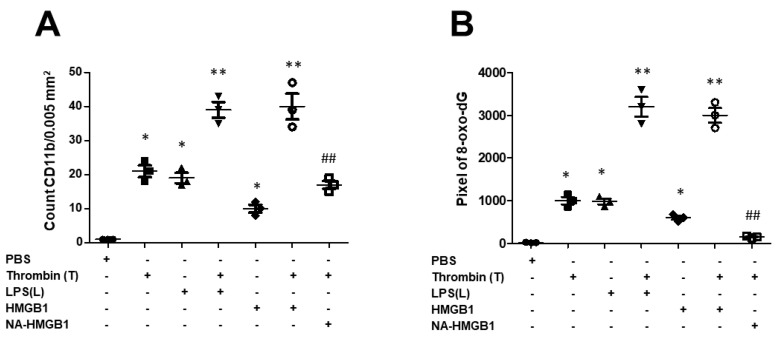
Quantitative analysis of oxidative stress of neuron and activated microglia cells. (**A**) Quantitative analysis of 8-oxo-dG expressed in pixel in different treatment groups. (**B**) Quantitative analysis of the number of IBA-1 in different treatment groups; N = 3, * *p* < 0.05; ** *p* < 0.01; ## *p* < 0.01 indicated T + NA-HMGB1 related to T + HMGB1.Black dot: PBS group; Black square: Thrombin(T) group; Back triangle: LPS (L) group; Inverted triangle: T+L group; Black diamond: HMGB1 group; Open circle: T + HMGB1; Open square: T + NA-HMGB1.

**Figure 5 ijms-24-12664-f005:**
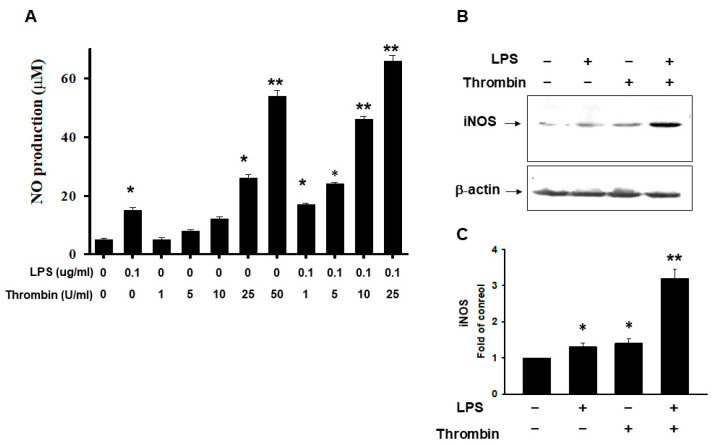
Release of NO and iNOS in microglia cells subjected to thrombin stimulation. (**A**) Microglial cells were incubated for 24 h with the indicated amount of thrombin (U/mL) and LPS. The amount of nitrite formed from NO was determined as described in the Methods section. Each value represents the mean ± SEM of three samples. (**B**) The representative of iNO expression detected by immunoblotting in microglia cells subjected to the stimulation by thrombin, LPS, or a combination of the two for the indicated times. (**C**) The quantitative analysis of immunoblotting of iNOS in microglia cells subjected to the stimulation by thrombin, LPS, or a combination of the two for the indicated times. Each value is the mean ± SEM of three samples. *: *p* < 0.05; ** *p* < 0.01, indicating a significant difference between experimental and control groups.

**Figure 6 ijms-24-12664-f006:**
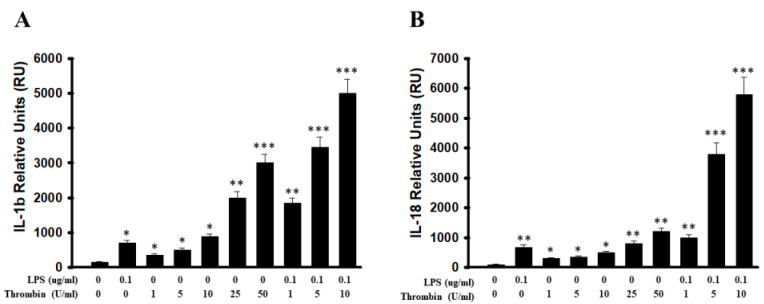
The expression of IL-1β and IL-18 mRNA in primary mouse brain microglia cells subjected to stimulation by thrombin, LPS, or the combination of the two. Primary microglia cells were stimulated with thrombin, LPS, or both thrombin and LPS for the 4 h, after which the total RNA was extracted, and cDNA was synthesized and subjected to analysis using real-time PCR. (**A**) The expression of IL-1β mRNA was qualitatively analyzed with escalating doses of thrombin from 1 to 50 units, and LPS served as a positive control. The expression of IL-1β mRNA was qualitatively analyzed in primary microglia subjected to combined escalating dosage of thrombin and LPS. (**B**) The expression of IL-18 mRNA was qualitatively analyzed with escalating doses of thrombin 1 to 50 units, and LPS served as a positive control. The expression of IL-18 mRNA was qualitatively analyzed in the primary microglia subjected to combined escalating dosage of thrombin and LPS. The data represent relative units (RU), that is, fold change in gene expression, which is normalized to an endogenous reference gene (18S rRNA) and is relative to no-template-control (NTC)-calibrator. Each microglia cell sample represents a pool of separately stimulated cells, and the results are representative of three independent but similarly performed experiments. Each value is the mean ± SEM of three samples. *: *p* < 0.05; **: *p* < 0.01; ***: *p* < 0.001, indicating a significant difference between experimental and control groups.

**Figure 7 ijms-24-12664-f007:**
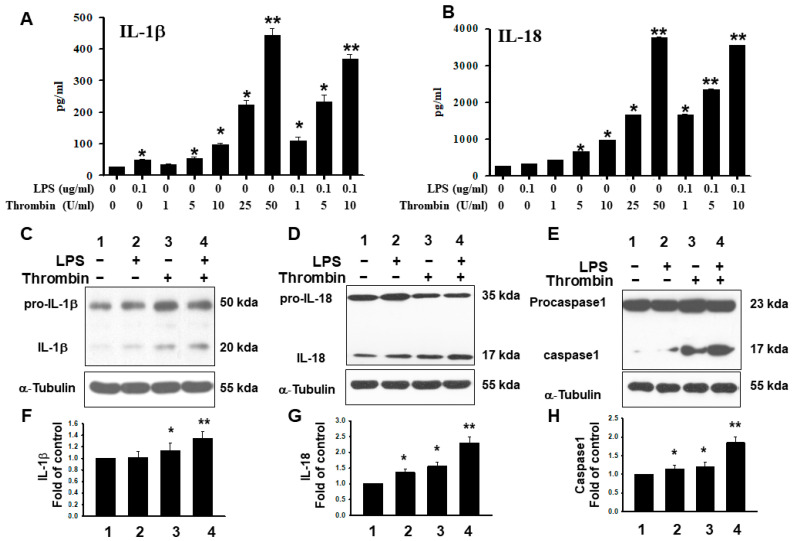
The increased activity of inflammasome in activated microglia subjected to thrombin and LPS stimulation. Microglia cells were activated with thrombin, LPS, or combination of the two for 24 h. IL-1β and IL-18 secretion from microglia cells was analyzed using ELISA. The total cellular protein lysates were prepared and subjected to Western blot analysis for pro-IL-1β, pro-IL-18, IL-1β, IL-18, and caspase I. (**A**) The quantitative analysis of IL-1β in microglia cells subjected to thrombin, LPS, or a combination of the two. (**B**) The quantitative analysis of IL-18 in microglia cells subjected to thrombin, LPS, or a combination of the two. (**C**) A representative Western blot analysis of pro-IL-1β, and IL-1β subjected to stimulation by thrombin, LPS, or a combination of the two. (**D**) A representative Western blot analysis in pro-IL-18, and IL-18 subjected to stimulation by thrombin, LPS, or the combination of the two. (**E**) A representative Western blot analysis of caspase I (P17 and P23) subjected to stimulation by thrombin, LPS, or a combination of the two. (**F**–**H**) The quantitative analysis of Western blot analysis in pro-IL-1β, pro-IL-18, IL-1β, IL-18, and caspase I was subjected to stimulation by thrombin, LPS, or a combination of the two. Each microglia cell sample represents a pool of separately stimulated cells. Each value is the mean ± SEM of three samples. *: *p* < 0.05; **: *p* < 0.01, indicating a significant difference between experimental and control groups.

**Figure 8 ijms-24-12664-f008:**
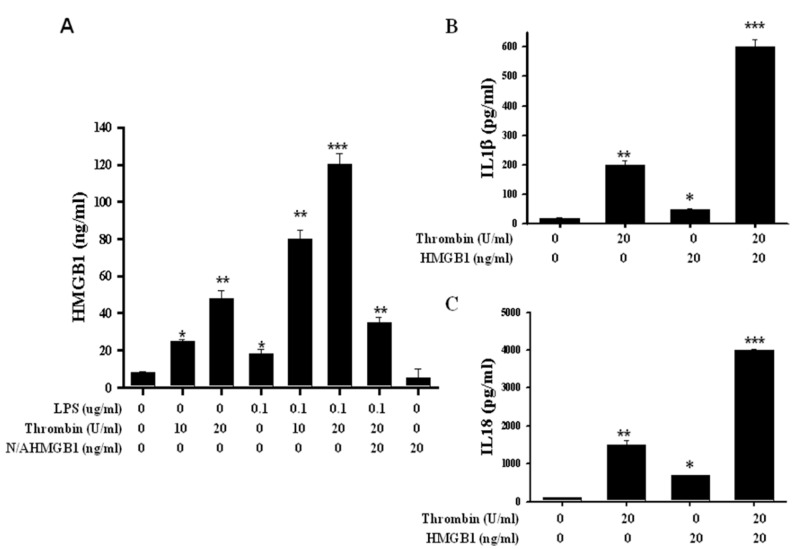
Release of HMGB1 in line with up-regulation of inflammasomes stimulated by thrombin and abolished by HMGB1 neutralized antibody. (**A**) The quantitative analysis of HGMB1 in supernatant in microglia subjected to thrombin or LPS stimulation as well as counteracted by Neutralize antibody HMGB1. Primary microglia cells were activated with thrombin or LPS for 18 h, after which total supernatant was synthesized for the determination of HMGB1. Neutralize antibody HMGB1 (NA-HMGB1, 20 μg/mL) was incubated in thrombin and/or LPS condition. (**B**) The quantitative analysis of the expression for IL-1β in microglia cells subjected to thrombin stimulation demonstrated an escalating response by adding HMGB1 recombinant protein. (**C**) The quantitative analysis in the expression of IL-18 in microglia cells subjected to thrombin stimulation and escalating response by adding HMGB1 recombinant protein. Each microglia cell sample represents a pool of separately stimulated cells. The results are representative of three independents, but similarly performed experiments. Values are means± SEM from three independent analyses. *: *p* < 0.05; **: *p* < 0.01; ***: *p* < 0.001.

**Figure 9 ijms-24-12664-f009:**
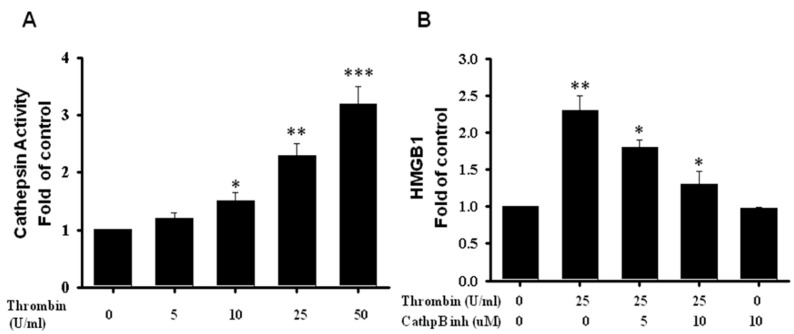
The expression of Cathepsin B activity related to HMGB1 expression when triggered by thrombin in primary mouse brain microglia cells. Primary microglia cells were stimulated with thrombin for 4 h. (**A**) The quantitative analysis of Cathepsin B activity related to an escalating thrombin dose. (**B**) The quantitative analysis of HMGB1 activity determined in microglia cells subjected to thrombin stimulation or combined with Cathepsin B inhibitors. Cathepsin B activity was measured via the degradation of a fluorescent substrate, Z-RR-AMC Cathepsin B Substrate. Magic Red Cathepsin B Substrate was added to microglia cells treated or not treated with thrombin. Thrombin-treated cells were also treated with 10 µM Ca-074-me, a specific inhibitor of cathepsin B. Degradation of the Z-RR-AMC Cathepsin B Substrate was measured via fluorescence ELISA reader. Values are means± SEM from three independent analyses. *: *p* < 0.05; **: *p* < 0.01; ***: *p* < 0.001. indicated the value related to the control group.

## Data Availability

Not applicable.

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
