# Peer review of "Neuronal Death Caused by HMGB1-Evoked via Inflammasomes from Thrombin-Activated Microglia Cells"

_ijms, 2023, doi:10.3390/ijms241612664_

Round 1
Reviewer 1 Report
The manuscript entitled 'Neuronal death caused by HMGB1 evoked via inflammasomes from thrombin-activated microglia cells' is a well-written piece of research that reports on the role the the HMGB-1 protein in the activation of inflammatory events leading to neuronal death. The topic is particularly relevant in the view of newly addressed international research on the role of immune system and inflammation in neurodegeneration.
To improve the readability, here are some remarks that authors should consider to address:
1. Introduction
- in line 64 indicate the meaning of the NOD acronym.
- be consistent with acronyms; in line 69, NALP3 appears as a new acronym compared with the previous lines: authors should substitute it with NLR family pyrin domain containing 3 (NLRP3) (whose last name was, in fact, NALP3);
- in line 73, DND binding protein: is it intentional (Dead-end (Dnd) RNA binding protein), or should it be substituted by DNA binding protein?
2. Results
- In line 281 the phrase 'lysosomal damages' should be changed to '... damage', unless there is evidence that cathepsin B creates damages of different type to the lysosomes.
- Line 282: NALP3, see comment above and change it consistently throughout the manuscript.
3. Discussion
- line 345: (PKR) can be removed since it is not mentioned anymore in the remaining text.
- lines 377-384: please refer to the CA1 and CA2 as CA1 sector and CA2 sector of the hippocampus proper, and not as 'areas'. In lines 381-382 the role as novelty detection of the CA3 sector may be not precise in conditions of stress. It was shown in fact that in the rat is the CA1 sector to to play a role as a ‘novelty’ detector, since it appears to identify the mismatches between the set of inputs from the entorhinal cortex (regarding the current situation), and those from the CA3 (related to the stored predictions (see Murakami et al., Neurosci. Res. 2005, 53, 129–139).
4 Materials and Methods
- par. 4.2., line 416-417: Please add information relating to species where antibodies were produced: by the way, the word antibody/ies is never mentioned in this sentence
- lines 426 and 439: remove the double dots from the subheading
- line 482: write 'um' in symbol character --> µm
- line 490: substitute 'for staining' with 'as secondary antibodies'
- lines 496-497: specify which images were analyzed with ImageJ and which ones with the UN-SCANT-IT gel TM software.
5. Figures
Figure 4: Regardless the used marker, it is not clear from the Y axis legend and from the figure legend what procedure has been used to measure: cell count or densitometry?
Figure 6: the graphs and legends are out of focus
Figure 7: The panels C and D have low resolution and photos appear to have been excessively adjusted for contrast. Authors should remake these panels to display high quality images. Please also upload the original blots in the editorial manager.
Author Response
The manuscript entitled 'Neuronal death caused by HMGB1 evoked via inflammasomes from thrombin-activated microglia cells' is a well-written piece of research that reports on the role the the HMGB-1 protein in the activation of inflammatory events leading to neuronal death. The topic is particularly relevant in the view of newly addressed international research on the role of immune system and inflammation in neurodegeneration.
To improve the readability, here are some remarks that authors should consider to address:
- Introduction
- in line 64 indicate the meaning of the NOD acronym.
Response to comments: Yes, it is. NOD should be “Nucleotide-binding oligomerization domain”. We make the correction.
- be consistent with acronyms; in line 69, NALP3 appears as a new acronym compared with the previous lines: authors should substitute it with NLR family pyrin domain containing 3 (NLRP3) (whose last name was, in fact, NALP3);
Response to comments: Yes, it is. It is our misspelling. It should be” NLR family pyrin domain containing 3 (NLRP3)”.
- in line 73, DND binding protein: is it intentional (Dead-end (Dnd) RNA binding protein), or should it be substituted by DNA binding protein?
Response to comments: Yes, it is. It is our misspelling. It should be” DNA binding protein”
- Results
- In line 281 the phrase 'lysosomal damages' should be changed to '... damage', unless there is evidence that cathepsin B creates damages of different type to the lysosomes.
Response to comments: Yes, it is. We should reduce the intonation of this sentence” Activation of cathepsin B through lysosomal damages contributes to the activation of NALP3 inflammasomes.” to” Cathepsin B Is required for inflammasome activation in immune cells through NLRP3 Interaction.”
- Line 282: NALP3, see comment above and change it consistently throughout the manuscript.
Response to comments: Yes, it is. It is our misspelling. We change all of NALP 3 to NLRP3.
- Discussion
- line 345: (PKR) can be removed since it is not mentioned anymore in the remaining text.
Response to comments: Yes, it is. We remove the abbreviation” PKR”.
- lines 377-384: please refer to the CA1 and CA2 as CA1 sector and CA2 sector of the hippocampus proper, and not as 'areas'. In lines 381-382 the role as novelty detection of the CA3 sector may be not precise in conditions of stress. It was shown in fact that in the rat is the CA1 sector to to play a role as a ‘novelty’ detector, since it appears to identify the mismatches between the set of inputs from the entorhinal cortex (regarding the current situation), and those from the CA3 (related to the stored predictions (see Murakami et al., Neurosci. Res. 2005, 53, 129–139).
Response to comments: Yes, it is. We revised this paragraph and add the new reference (Murakami et al., Neurosci. Res. 2005, 53, 129–139), which made the manuscript much more readable.
In line 381-381, we revised as below “CA3 sector supports processes involved in spatial pattern association, spatial pattern completion, novelty detection, and short-term memory. The CA1 area is responsible for the temporal pattern completion and intermediate-term memory [70]. In some aspects, the CA1 sector plays a role as a ‘novelty’ detector, since it appears to identify the mismatches between the set of inputs from the entorhinal cortex (regarding the current situation), and those from the CA3 [75].
4 Materials and Methods
- par. 4.2., line 416-417: Please add information relating to species where antibodies were produced: by the way, the word antibody/ies is never mentioned in this sentence
Response to comments: We added species of antibodies and put the word” antibodies in the sentences “ -- Blots were incubated for 1 h with the iNOS (# 06-573; Rabbit Polyclonal; 1:200; Merck Mil-lipore), caspase-I (# No:bs-0169R; Rabbit Polyclonal;1:200; Bioss) and β actin (# A2228; Mouse Monoclonal; :200; Sigma-Aldrich) antibodies”.
- lines 426 and 439: remove the double dots from the subheading
Response to comments: Yes, it is. We remove the double dots.
- line 482: write 'um' in symbol character --> µm
Response to comments: Yes, it is. We revised” um” to “μm’.
- line 490: substitute 'for staining' with 'as secondary antibodies'
Response to comments: Yes, it is. We replaced “ for staining” to secondary antibodies”.
- lines 496-497: specify which images were analyzed with ImageJ and which ones with the UN-SCANT-IT gel TM software.
Response to comments: Yes, it is. We revised this section” –The numbers of cells (CD11b positive) were counted in 20 square areas randomly selected from a 100-square ocular grid. Six samples in each immunohistochemical staining were analyzed. Fluorescent imaging was performed on an Olympus BX40 Research Microscope, using the same laser power and exposure time. Images were quantitatively analyzed by the software ImageJ and UN-SCAN-IT gelTM (Gel & Graph Digitiz-ing Software Version 6.1). The counts of CD11b were analyzed by UN-SCANT-IT gel TM software and the others were analyzed by image J software.”
- Figures
Figure 4: Regardless the used marker, it is not clear from the Y axis legend and from the figure legend what procedure has been used to measure: cell count or densitometry?
Response to comments: Yes, it is. We revised the calculation method in the methodology” The numbers of cells (CD11b positive) were counted in 20 square areas randomly selected from a 100-square ocular grid. Six samples in each immunohistochemical staining were analyzed. Fluorescent imaging was performed on an Olympus BX40 Research Microscope, using the same laser power and exposure time. Images were quantitatively analyzed by the software ImageJ and UN-SCAN-IT gelTM (Gel & Graph Digitiz-ing Software Version 6.1). The counts of CD11b were analyzed by UN-SCANT-IT gel TM software and the others were analyzed by image J software.”
Figure 6: the graphs and legends are out of focus
Response to comments: Yes, it is. We condensed four figures to two figures to make them readable.
Figure 7: The panels C and D have low resolution and photos appear to have been excessively adjusted for contrast. Authors should remake these panels to display high quality images. Please also upload the original blots in the editorial manager.
Response to comments: Yes, it is. We repeated Figure 7 to have high resolution imaging.
Reviewer 2 Report
1. Fig4. Please provide detailed plots. As ancillary information authors should provide the information on their criteria of considering pixels for their analysis in the method section.
2. E.g., in Fig.6. why separate scaling metrics were used to compare the relative units, should be uniform at least across the figure.
3. The figures representing the behavioral test quantification are hard to follow, please make them at least visible so the comparison and conclusion drawn are easily depicted.
4. Colocalization is not a physiological measure of thrombin activity in context to inflammasome components. Authors are advised to provide more direct physiological evidence.
5. It is advised to characterize the individual effects of LPS and thrombin on cell viability along with their synergistic effects to better under understand the control conditions.
6. Fig.7. blots should be replaced, the quantifications done from these blots aren’t specific.
7. Section 2.5, after line 286, the manuscript is missing or discontinued, please check, and provide the complete section.
8. Please revisit the manuscript for typos, grammatical errors and enhance the figure quality if possible.
Sections of the manuscript are missing and many sentences needs to be rephrased or rebuilt for the respective clarity.
Author Response
- Please provide detailed plots. As ancillary information authors should provide the information on their criteria of considering pixels for their analysis in the method section.
Response to comments: Yes, it is. We revised the methodology to make them more readable and understood. “The numbers of cells (CD11b positive) were counted in 20 square areas randomly selected from a 100-square ocular grid. Six samples in each immunohistochemical staining were analyzed. Fluorescent imaging was performed on an Olympus BX40 Research Microscope, using the same laser power and exposure time. Images were quantitatively analyzed by the software ImageJ and UN-SCAN-IT gelTM (Gel & Graph Digitiz-ing Software Version 6.1). The counts of CD11b were analyzed by UN-SCANT-IT gel TM software and the others were analyzed by image J software.”
- g., in Fig.6. why separate scaling metrics were used to compare the relative units, should be uniform at least across the figure.
Response to comments: Yes, it is. We unify the relative units in all figures. We revised this section and re-edited the figure as follow:” We found a very mild response of IL-1 β after stimulation by thrombin at the dosage of 5 to 10 units. LPS synergistically increased mRNA expressions of IL-1β in microglia cells following low dosage of thrombin from 5 to 10 units (Figure 6A). The dosage of thrombin >10 units exerted the marked expression of IL-18 mRNA with escalated response up to 50 units, and LPS was used as a positive control. We also found a very mild response of IL-18 after stimulation by thrombin at the dosage of 5 to 10 units. LPS synergistically increased mRNA expressions of IL-18 in microglia cells following low dosage of thrombin from 5 to 10 units (Figure 6B)”.
- The figures representing the behavioral test quantification are hard to follow, please make them at least visible so the comparison and conclusion drawn are easily depicted.
Response to comments: Yes, it is. We make the additional bar graph in the different time points to make them readable shown in Supplementary Figure 1S. We added one paragraph “ For more details on the comparison in the different groups related to the different time profile, the plots illustrated in bar graph were shown in Supplementary Figure 1 (Figure 1S).”
- Colocalization is not a physiological measure of thrombin activity in context to inflammasome components. Authors are advised to provide more direct physiological evidence.
Response to comments: Yes, it is. We add the western blot and quantify the confocal imaging shown in Figure 2S. We add one paragraph “ The obtained tissue in the region of hippocampus were obtained for the western blot analysis and the intensity of immunohistochemistry staining were also quantified ( Supplementary Figure 2S).”
- It is advised to characterize the individual effects of LPS and thrombin on cell viability along with their synergistic effects to better under understand the control conditions.
Response to comments: We added the MTT assay in the cell culture experiment in Figure 3S. We add one paragraph “The cell morphology alteration and cell viability were shown in Supplementary Figure 3S. The data of MTT assay approach the expression of NO production.”
- 7. blots should be replaced, the quantifications done from these blots aren’t specific.
Response to comments: Yes, it is. We repeated the same experiment to make them readable and more understood and add the original data in Figure 4S.
- Section 2.5, after line 286, the manuscript is missing or discontinued, please check, and provide the complete section.
Response to comments: Yes, it is. We rewrite to make them into a complete section.
- Please revisit the manuscript for typos, grammatical errors and enhance the figure quality if possible.
Response to comments: Yes, it is. Dr. Jason Sheehan, a native English researcher, had to re-edit English for correcting the typos and grammatical errors.
Round 2
Reviewer 1 Report
The manuscript has fairly improved since the authors have accomplished the received suggestions. However, there are still some remarks, due to incorrect phrasing, or repetitions of sentences, or unclear statements.
Examples are:
lines 155-159
"In further quantitative analysis in the number of activated microglia cells and 8-oxo-dG, results further confirmed the immunohistochemical findings (Figure 4A, B). The obtained tissue in the region of hippocampus were subjected for the western blot analysis. The intensity of immunohistochemistry staining was also quantified (Supplementary Figure 2S), which paralleled the above-said results."
where the last sentence should stay matched with the first one.
lines 184-186
"Larger expressions of iNOS were found in microglia cells subjected to the treatment of thrombin or LPS. LPS exerted a synergistic effect on thrombin in augmenting iNOS expressions (Figure 5B, C)." It appears the two sentences express in different ways the same finding.
lines 232-238:
"The expression of IL-1 mRNA was qualitatively analyzed in primary microglia subjected to combined escalating dosage of thrombin and LPS.(B) The expression of IL-18 mRNA was qualitatively analyzed in the escalated dosage from 1 to 50 units of thrombin, and LPS served as a positive control. The expression of IL-18 mRNA was qualitatively analyzed in the primary microglia subjected to combined escalating dosage of thrombin and LPS."
though speaking of different ILs, maybe the text should be still optimized.
lines 513-516:
"Images were quantitatively analyzed by the software ImageJ and UN-SCAN-IT gelTM (Gel & Graph Digitizing Software Version 6.1). The counts of CD11b were analyzed by UN-SCANT-IT gel TM software and the others were analyzed by image J software." Text can be optimized.
line 506: "were used to secondary" is grammatically incorrect.
The adds on the authors made are fine but sometimes there is no matching between the newly added text and the previous one, instead in several cases a whole revision to the sentences should have been done. The quality of English is poor.
Author Response
The manuscript has fairly improved since the authors have accomplished the received suggestions. However, there are still some remarks, due to incorrect phrasing, or repetitions of sentences, or unclear statements.
Examples are:
lines 155-159
"In further quantitative analysis in the number of activated microglia cells and 8-oxo-dG, results further confirmed the immunohistochemical findings (Figure 4A, B). The obtained tissue in the region of hippocampus were subjected for the western blot analysis. The intensity of immunohistochemistry staining was also quantified (Supplementary Figure 2S), which paralleled the above-said results."
where the last sentence should stay matched with the first one.
Response to comments: Yes, it is. We revised this paragraph shown below” The obtained tissue in the region of hippocampus were subjected for the western blot analysis. The intensity of immunohistochemistry staining was also quantified (Supplementary Figure 2S), and the analysis showed the same expression pattern consistent with the representative western blots.”
lines 184-186
"Larger expressions of iNOS were found in microglia cells subjected to the treatment of thrombin or LPS. LPS exerted a synergistic effect on thrombin in augmenting iNOS expressions (Figure 5B, C)." It appears the two sentences express in different ways the same finding.
Response to comments: Yes, it is. We edited it as below” The iNOS were induced in microglia cells subjected to either thrombin or LPS treatment. However, LPS exerted a synergistic effect on thrombin in augmenting iNOS expressions (Figure 5B, C).”.
lines 232-238:
"The expression of IL-1b mRNA was qualitatively analyzed in primary microglia subjected to combined escalating dosage of thrombin and LPS.(B) The expression of IL-18 mRNA was qualitatively analyzed in the escalated dosage from 1 to 50 units of thrombin, and LPS served as a positive control. The expression of IL-18 mRNA was qualitatively analyzed in the primary microglia subjected to combined escalating dosage of thrombin and LPS."
though speaking of different ILs, maybe the text should be still optimized.
Response to comments: Yes, it is. We rewrote this paragraph “To investigate the involvement of inflammasomes in the above thrombin activation, we determined mRNA expressions of IL-1β and IL-18 in microglia cells. Thrombin had activated a marked expression of IL-1β with a progressive increase in this response from 1 to 50 units thrombin, whereas LPS served as the positive control. LPS synergistically increased mRNA expressions of IL-1β in microglia cells following the dosage of thrombin from 1 to 10 units (Figure 6A). The escalated dosage of thrombin from 1 to 50 units exerted the significant expression of IL-18 mRNA, and LPS was used as a positive control. LPS synergistically increased mRNA expressions of IL-18 in microglia cells following the dosage of thrombin from 1 to 10 units (Figure 6B).”
lines 513-516:
"Images were quantitatively analyzed by the software ImageJ and UN-SCAN-IT gelTM (Gel & Graph Digitizing Software Version 6.1). The counts of CD11b were analyzed by UN-SCANT-IT gel TM software and the others were analyzed by image J software." Text can be optimized.
Response to comments: We add one paragraph as below:” In brief, the imaging of immunohistochemistry staining was imported. Next, the original image is then split into 3 single colored image and the unwanted image was closed. The desired images were saves as TIFF files. In the threshold window, pixel that are within the red box were selected and the dark background was also selected. The bar was dragged as needed to modify the selected area. A binary image was created and fluorescence intensity and signal were determined. All the stained pixels were converted to black, and all the non-stained pixels were converted to white. The number of black pixels then indicated the area showing positive staining. The data were shown in the number of back pixel/mm2.”
line 506: "were used to secondary" is grammatically incorrect.
Response to comments: Yes, it is. It is our mistake and we corrected them as “-- were used for secondary---".
The adds on the authors made are fine but sometimes there is no matching between the newly added text and the previous one, instead in several cases a whole revision to the sentences should have been done. The quality of English is poor.
Response to comments: Yes, it is. The final version was corrected again by Dr. Jason Sheehan and marked with green color.
Reviewer 2 Report
1. Line 48-49; there are grammatical errors.
2. Findings mentioned in lines 54-55, needs appropriate citations.
3. Lines 57-58 pargraph-2 of the introduction, there is no smooth transition to introduce thrombin, why thrombin suddenly?
4. Same for the paragraph and rest of the introduction section.
5. What are the major downstream pathways affected in various nervous system cells types?
6. What is the key contributing molecular mechanisms that are contributing to cell death and are inflammasome mediated in your model.
7. What are the downstream physiological impacts of such an injection or Ab-injection in your model?
Need-work.
Author Response
Line 48-49; there are grammatical errors.
Response to comments: Yes, it is. We rewrote this paragraph” Neuroinflammation has been implicated in neurodegenerative disorders for decades. However, the exactly timing and mechanism of inflammation process contributing to neurodegenerative disorders remains poorly understood [1].
Findings mentioned in lines 54-55, needs appropriate citations.
Response to comments: Yes, it is. We added two references at the end of this paragraph:
Pan, H. C.; Yang, C. N.; Hung, Y. W.; Lee, W. J.; Tien, H. R.; Shen, C. C.; Sheehan, J.; Chou, C. T.; Sheu, M. L., Reciprocal modulation of C/EBP-α and C/EBP-β by IL-13 in activated microglia prevents neuronal death. Eur J Immunol 2013, 43, 2854-65.
Sheu, M.L.; Pan, L.Y.; Yang, C. N.; Sheehan J.; Pan, L.Y.; You, W.C.; Wang, C.C.; Pan, H.C., Thrombin-induced microglia activation modulated through aryl hydrocarbon receptors. Int. J. Mol. Sci 2023, 24,11416. https://doi.org/10.3390/ijms241411416
Lines 57-58 pargraph-2 of the introduction, there is no smooth transition to introduce thrombin, why thrombin suddenly?
Response to comments: Yes, it is. We revised this paragraph to make them smoothly and readably. We rewrote this paragraph “Recently, mounting evidence showed that certain elements of the coagulation cascade initiated and propagated the inflammation response in the central nervous system. The potentially important protein related to both coagulation and inflammation is thrombin [78,79]. ---”
De, Luca C.; Virtuoso, A.; Maggio, N.; Papa, M., Neuro-Coagulopathy: Blood Coagulation Factors in Central Nervous System Diseases. Int J Mol Sci 2017, 18.
Göbel, K.; Eichler, S.; Wiendl, H.; Chavakis, T.; Kleinschnitz, C.; Meuth, S. G., The Coagulation Factors Fibrinogen, Thrombin, and Factor XII in Inflammatory Disorders-A Systematic Review. Front Immunol 2018, 9, 1731.
Same for the paragraph and rest of the introduction section.
Response to comments: Yes, it is. The final version was re-edited and corrected by our team member, Dr. Jason Sheehan and the revision in any change was marked with green color.
What are the major downstream pathways affected in various nervous system cells types?
Response to comments: In our previous study, thrombin exerted the widely spreading effects in the central nervous system involved in the microglia activation, disruption of endothelium related to blood brain barrier breakdown as well as the direct effect on neuronal survival [77]. In the current study, we only focus on thrombin-treated microglia to show a downstream activation of inflammasomes thereby contributing to HMGB1 secretion. The thrombin-activated expressions of IL-1 β and IL-18, coupled with the activation of caspase-I, were augmented by HMGB1 but abolished by anti-HMGB1 antibody. These effects were highly correlated with changes in neurobehavior and neuronal survival. Hence, modulating HMGB1 in the thrombin-activated inflammatory responses can play a potential role in rescuing from a neurodegenerative disorder.
What is the key contributing molecular mechanisms that are contributing to cell death and are inflammasome mediated in your model.
Response to comments: In our data, thrombin activated expressions of IL-1 β and IL-18, coupled with the activation of caspase-I, which accelerated the HMGB1 secretion to induce the neuronal death.
What are the downstream physiological impacts of such an injection or Ab-injection in your model?
Response to comments: Base on the hypothesis, thrombin activated microglia cells to trigger the inflammasome complex reaction (increased IL-1b,IL-18, and caspase I) and then accelerated the secretion of HMGB1, which contributed to neuronal death. In this study, for the test of the effect of HMBG1 on microglia cells, the thrombin activated microglia cells were manipulated by HMGB1 and its neutralized antibody to validate our hypothesis.
Comments on the quality of English language: Nee to work
Response to comments: The manuscript in English were re-edited by Jason Sheehan, who is our research member and also a professor in the department of neurosurgery, University of Virginia, USA.
Round 3
Reviewer 2 Report
1. “Microglia, the tissue macrophages of the brain, under healthy conditions are a resting phenotype characterized by a ramified morphology, and they extend their fine processes to scan the environment.” Where is the reference supporting this and what kind of glial-processes are in picture? This was pointed out in the earlier review.
2. Please show individual datapoints instead of bar diagrams, so an overview of data distribution can be seen.
3. Fig.7.D, the band representing IL-18 secretion, condition-1 (without LPS and thrombin), was it consistent in all the repetitions, if it was reconfirmed? If not, then please do so?
4. Does thrombin globally activate the Cathepsin-B or it is restricted to a specific cellular compartment?
5. Cathepsins are also involved with secretion of vesicles that are of endosomal origin, please review the literature and modify the manuscript accordingly.
6. Were their any changes to brain vasculature of the animals used in these experiments or do author expect some effects to vasculature?
7. Please pay attention to the conclusion section.
There are still errors in manuscript in terms of grammar and typos etc. please review.
Author Response
Comments and Suggestions for Authors
- “Microglia, the tissue macrophages of the brain, under healthy conditions are a resting phenotype characterized by a ramified morphology, and they extend their fine processes to scan the environment.” Where is the reference supporting this and what kind of glial-processes are in picture? This was pointed out in the earlier review.
Response to comments: We re-edited this paragraph “Microglia, the tissue macrophages of the brain, under healthy conditions are a resting phenotype characterized by a ramified morphology, and they extend their fine processes to scan the environment [80,81]. In response to a homeostatic disturbance, microglia rapidly change their phenotype, thereby contributing to processes such as inflammation, stroke, trauma, tissue remodeling, and neurogenesis [4-6, 71, 77].
- Eggen, B. J.; Raj, D.; Hanisch, U. K.; Boddeke HW.; Microglial phenotype and adaptation. J Neuroimmune Pharmacol 2013, 8, 807-23.
- Ni, J.; Lan, F.; Xu, Y.; Nakanishi, H.; Li, X.;. Extralysosomal cathepsin B in central nervous system: Mechanisms and therapeutic implications. Brain Pathol 2022, 32, e13071.
- Please show individual datapoints instead of bar diagrams, so an overview of data distribution can be seen.
Response to comments: We try to replace our bar diagram with individual datapoints from Figure 4 to 9. But this presentation was not as good as the previous one. We only replaced Fig 4 with individual datapoints.
- Fig.7.D, the band representing IL-18 secretion, condition-1 (without LPS and thrombin), was it consistent in all the repetitions, if it was reconfirmed? If not, then please do so?
Response to comments: In Figure 7D, we conducted three independent experiments, and the results showed the same trends.
- Does thrombin globally activate the Cathepsin-B or it is restricted to a specific cellular compartment?
Response to comments: In a large volume of studies concerning the LPS induced microglia activation, LPS trigger inflammasome reaction vial NLRP3 which contributed to release of HMGB1. Interestingly, a diverse set of NLRP3 inflammasome activators have been used to show the requirement of Cathepsin B for IL-1β production. Cathepsin B also involves in the activation of NFκB, which is responsible for the transcriptional regulation of NLRP3 and IL-1β.
Up to date, there was few studies to investigate to role of thrombin involved in microglia activation through this pathway. For the further verification of this possibility, we determine the Cathepsin activity as an intermediate factor to assure the cellular response to thrombin to accelerate HMGB1 activity. We found that thrombin trigger Cathepsins activity and the paralleling increase in HMGb1 secretion is attenuated by Cathepsin inhibitor, which further approve the involvement of inflammasome in this pathway. Considering your comments on globally or specific cellular compartment involved in Cathepsin B, it is not our aim of this study. The issue was very interesting and we hope to investigate in the near future.
- Cathepsins are also involved with secretion of vesicles that are of endosomal origin, please review the literature and modify the manuscript accordingly.
Response to comments: We modify the manuscript according to our comments: “---The mannose-6 phosphate and mannose-6 phosphate receptor-mediated transport of pro-cathepsin from the trans-Golgi network to endo/lysosome. In lysosomes, pro-Cathepsin further processed via autocatalysis into a mature two-chain form composed of an N-terminal light chain and a C-terminal heavy chain [82, 83].
- Olah, M.; Biber, K.; Vinet, J.; Boddeke, HW.; Microglia phenotype diversity. CNS Neurol Disord Drug Targets, 2011, 10, 108-18.
- Xie, Z.; Zhao, M.; Yan, C.; Kong, W.; Lan, F.; Narengaowa.; Zhao, S.; Yang, Q.; Bai, Z.; Qing, H.; Ni, J.; Cathepsin B in programmed cell death machinery: mechanisms of execution and regulatory pathways. Cell Death Dis 2023, 14, 255.
- Were their any changes to brain vasculature of the animals used in these experiments or do author expect some effects to vasculature?
Response to comments: High mobility group box-1 (HMGB1), a nonhistone nuclear protein, is ubiquitously expressed in almost all kinds of cells. At our previous study, thrombin exerted the widely spreading effects in the central nervous system involved in the microglia activation, disruption of endothelium related to blood brain barrier breakdown as well as the direct effect on neuronal survival. As in the other report, Intravenous injection of anti-HMGB1 mAb significantly protected against BBB disruption induced by ischemia and hemorrhage in rats. As expected, HMGB1 may contribute to some sort of effect on vasculature structure. Our present study focused on the mechanism of thrombin triggered microglia reaction in the release of HMGB1. In the animal’s study, we only measured the neuronal response to HMGB1 release from the thrombin treated microglia cells.
- Please pay attention to the conclusion section.
Response to comments: Yes, it is. We reduce the intonation and rewrite as below: Thrombin may induce microglia activation triggered inflammasomes to produce the down-stream of HMGB-1 and resulted in detrimental effects to nearby neurons. Such effects were attenuated by HMGB1 neutralizing antibodies. The refinement of HMGB-1 expression may be a useful tool for modulating neuro-inflammatory responses and thereby attenuating a thrombin associated central nervous system degenerative disorder cascade.